# Metabolic Predictors of Equine Performance in Endurance Racing

**DOI:** 10.3390/metabo11020082

**Published:** 2021-01-31

**Authors:** Anna Halama, Joao M. Oliveira, Silvio A. Filho, Muhammad Qasim, Iman W. Achkar, Sarah Johnson, Karsten Suhre, Tatiana Vinardell

**Affiliations:** 1Department of Physiology and Biophysics, Weill Cornell Medicine-Qatar, Doha 24144, Qatar; iwa2004@qatar-med.cornell.edu; 2Equine Veterinary Medical Center, Qatar Foundation, Doha 5825, Qatar; joaooliveiravet@gmail.com (J.M.O.); mqasim@qf.org.qa (M.Q.); sjohnson@qf.org.qa (S.J.); 3Department of Endurance Racing, Al Shaqab, Doha 36623, Qatar; sfilho@qf.org.qa; 4College of Health and Life Sciences, Hamad Bin Khalifa University, Member of Qatar Foundation, Doha 34110, Qatar

**Keywords:** endurance race, Arabian horses, metabolomics, fatty acid oxidation, amino acid metabolism

## Abstract

Equine performance in endurance racing depends on the interplay between physiological and metabolic processes. However, there is currently no parameter for estimating the readiness of animals for competition. Our objectives were to provide an in-depth characterization of metabolic consequences of endurance racing and to establish a metabolic performance profile for those animals. We monitored metabolite composition, using a broad non-targeted metabolomics approach, in blood plasma samples from 47 Arabian horses participating in endurance races. The samples were collected before and after the competition and a total of 792 metabolites were measured. We found significant alterations between before and after the race in 417 molecules involved in lipids and amino acid metabolism. Further, even before the race starts, we found metabolic differences between animals who completed the race and those who did not. We identified a set of six metabolite predictors (imidazole propionate, pipecolate, ethylmalonate, 2R-3R-dihydroxybutyrate, β-hydroxy-isovalerate and X-25455) of animal performance in endurance competition; the resulting model had an area under a receiver operating characteristic (AUC) of 0.92 (95% CI: 0.85–0.98). This study provides an in-depth characterization of metabolic alterations driven by endurance races in equines. Furthermore, we showed the feasibility of identifying potential metabolic signatures as predictors of animal performance in endurance competition.

## 1. Introduction

Endurance sport refers to any kind of competition in which participants’ stamina is tested to its limits. The performance of endurance athletes depends on multiple components, including genetic predisposition, physiological, biomechanical, and psychological factors [1]. The endurance performance capacity could be maximized with training, enabling the utilization of the individual’s genetic potential [2].

Endurance horses experience similar levels of physiological and metabolic stress as human athletes, and thus could be considered as a relevant in vivo model for the monitoring and establishment of a metabolic performance profile. For many horses, endurance racing is challenging and between 30 and 70% of animals are eliminated due to various health conditions, such as lameness and metabolic disturbances, which greatly correlate with environmental factors, riders experience and the horse’s level of fitness [3]. Therefore, strategies which could estimate the animal’s readiness to participate in endurance racing would be of great value, as this may reduce the number of unprepared animals resulting in a lower risk of developing certain health complications. However, currently there is no method that could estimate the equine endurance capacity.

The technological advancement in mass spectrometry (MS) contributed to the development of metabolomics, which offers detailed description of the small molecule (metabolite) composition of the organism [4]. Implementation of metabolomics approaches, in the field of endurance exercises resulted in comprehensive characterization of altered metabolic processes as a result of undergoing endurance performance [4,5,6,7]. Metabolic shift from carbohydrate towards lipid catabolism was suggested as the main energy source during endurance challenge in humans [5,6] as well as horses [8,9,10,11]. Additionally, amino acid catabolism and hemoglobin metabolism [5,12], as well as oxidative stress [6], were found to be triggered by such exercises. Metabolic pathways that are predominantly affected by endurance training could be further analyzed to define individual metabolic fitness and capacity to endurance sport exposure. Nevertheless, metabolic signatures indicating performance capacity and conditioning for endurance training remain elusive.

The aim of our study was to provide an in-depth characterization of metabolic consequences of endurance racing in Arabian and half-Arabian horses. Furthermore, we aimed to establish a metabolic performance profile for those animals which could serve as a proxy for the animal’s readiness to participate in an endurance race. To achieve this, we monitored blood plasma samples of selected animals before and after completing endurance competition, as well those eliminated due to lameness or metabolic conditions, using an MS-based broad untargeted metabolic profiling method. Notably, previous studies have so far only been performed in moderate temperate regions and only using more limited NMR methods for metabolic measurements [11,13]. Our study is the first of its kind to be performed under desert conditions and to deploy broad untargeted metabolic profiling.

## 2. Results

### 2.1. Equine Physiological Characteristic

A total of 47 horses were included in the study. The experimental design is presented in Figure 1A. Every horse participated in at least one endurance race of 80, 100 or 120 km distance, and some horses underwent several races. Overall, 34 horses raced once, 11 horses raced twice, and 2 horses raced three times. We obtained a total of 124 samples, including 62 samples from before and 62 samples from after the race. 

Out of the 62 racing horses, only 22 successfully finished the race, the remaining 40 were eliminated due to metabolic conditions (15 animals), determined by a heart rate auscultation above 64 beats per minute after the regulated rest period, low gastrointestinal motility or respiratory disturbances, lameness (22 animals) characterized by an abnormal gait of the animal at walk or trot, and other causes, such as falling of the rider (3 animals) (Appendix A). The actual distance achieved by each animal is provided in Appendix A.

Next, we tested whether physiological and clinical chemistry parameters monitored before the race could predict the outcome of the competition. The age was not significantly different between the animals finishing the race and those eliminated. The heart rate as well as all clinical chemistry parameters monitored before the race were not significantly different between the animals completing the race and those eliminated due to the metabolic conditions or lameness (Table 1). After the race, we observed significant increase in the heart rate as well as levels of albumin (ALB), total bilirubin (BIL), lactate (LACT), and lactate dehydrogenase (LDH), along with a significant decrease in blood urea nitrogen (BUN) between the animals completing the race and those that were eliminated (Table 1).

We also tested for dehydration, which could affect animal performance and result in elimination. We measured plasma osmolality, as a proxy for animal dehydration [14]. There were no significant differences in the osmolality levels between finishers and eliminated animals before the race (Appendix A) as well as after the race (Appendix A), suggesting that the performance was not affected by the animal hydration. Additionally, no significant changes in the osmolality levels before and after the race were observed across animals participating in all distance (80 km, 100 km, 120 km) races (Appendix A).

Taken together, this indicates that the parameters commonly used to monitor the equine health status prior to the competition are insufficient to determine the animal’s readiness to compete in an endurance race.

### 2.2. Endurance Race Triggers Metabolic Shift in Equine Metabolism 

In previous studies, enhanced lipid catabolism and protein degradation were reported as metabolic signatures of endurance racing [10,13,15]. However, those studies used an NMR-based metabolomics approach, which provides only a narrow range of metabolite coverage. Given that the number of metabolites which could be detected using NMR is limited in comparison to LC/MS based technologies [16], we used the broad non-targeted metabolomics platform from Metabolon Inc., using four complementary UPLC/MS runs to provide further insight into the equine physiology related to endurance performance.

We quantified relative levels of 792 metabolites, including 659 molecules of known identity, and 133 molecules of unknown identity. The distribution of measured metabolites across different metabolic classes is presented in Figure 1B and all measured metabolites are listed in Appendix A. The molecules of known identity cover eight primary pathways related to the metabolism of amino acids (183 molecules), carbohydrates (22 molecules), cofactors and vitamins (21 molecules), energy (10 molecules), lipids (265 molecules), nucleotides (28 molecules), peptides (29 molecules), and xenobiotics (101 molecules).

Principal component analysis (PCA) was conducted on the metabolite levels identified from those horses that were not eliminated and finished the competition. We found a clear separation into two groups between before and after the race (Figure 1C) suggesting metabolic alterations in response to the endurance competition. We observed a tight clustering of the samples before the race, while samples were more spread-out after the race. The relatively loose clustering of the samples observed after the race was not driven by the differences in the distance (Appendix A), and thus might suggest unique metabolic responses to endurance racing for each individual animal. The corresponding loading plot (Figure 1D) allowed for identification of those metabolites that contributed the most to the separation between the groups. The loading plot suggests that the molecules involved, predominantly in lipids and amino acid metabolism, contribute to this separation. 

### 2.3. Catabolism of Amino Acids and Lipids as well as Lactate Production Are Enhanced by Endurance Racing

In those animals that finished the race, we identified 417 metabolites, which were significantly altered after the race at a stringent Bonferroni level of significance (correction for number of metabolites (0.05/792), *p*-value = 6.3 × 10^−5^) (Appendix A). All monitored pathways were found to be affected by endurance racing, with the strongest alterations in lipids (167 out of 265 measured molecules), amino acids (106 out of 183 measured molecules), and carbohydrates (14 out of 22 measured molecules).

Among the lipid compounds, we found an increase in 103 molecules including free long chain fatty acids (saturated and unsaturated), dicarboxylic acids, and acylcarnitine with various chain lengths, as well as a decrease in the levels of 64 molecules—mainly lysophosphatidylcholines, phosphatidylcholines, phosphatidylethanolamines, and sphingomyelins. The observed lipid alterations suggest an accelerated lipid catabolism and utilization of fatty acids for energy production in different processes of fatty acid oxidation. For instance, elevated levels of free fatty acids and acylcarnitines with even or odd chains of fatty acids suggest increased β-oxidation, and α-oxidation, respectively, whereas accumulation of dicarboxylic acids suggests an accelerated process of ω-oxidation. Examples of the molecules involved in β-oxidation (Figure 2A), α-oxidation (Figure 2B), and ω-oxidation (Figure 2C) are presented in Figure 2.

We found that endurance racing also has an impact on multiple pathways of amino acid metabolism, including molecules involved in branch chain amino acid (BCAA), aromatic amino acid (AAA), histidine, lysine, methionine and urea cycle.

A significant decrease in valine (*p*-value = 1.15 × 10^−8^) and isoleucine (*p*-value = 1.12 × 10^−12^), together with the significant increase in their products of catabolism (e.g., 3-methyl-2-oxobutyrate (*p*-value = 1.49 × 10^−19^), 3-hydroxyisobutyrate (*p*-value = 4.35 × 10^−20^), and 3-methyl-2-oxovalerate (*p*-value = 2.84 × 10^−12^)), suggest activation of BCAA catabolism, potentially to support the energetic needs triggered by endurance racing. 

The levels of AAA, including phenylalanine (*p*-value = 1.65 × 10^−6^) and tryptophan (*p*-value = 4.37 × 10^−11^), were significantly decreased after the race and multiple products of their metabolism including N-acetylphenylalanine (*p*-value = 2.18 × 10^−5^), N-acetyltyrosine (*p*-value = 5.77 × 10^−10^), kynurenate (*p*-value = 2.15 × 10^−14^), and picolinate (*p*-value = 1.81 × 10^−5^) were elevated. We also observed significant alterations in histidine metabolism, including decrease in histidine levels together with increase in carnosine and anserine as well as 4-imidazoleacetate, imidazole lactate and formiminoglutamate (Figure 3). Among the molecules involved in lysine metabolism, we observed significant decrease in levels of pipecolate (*p*-value = 1.41 × 10^−17^) together with an increase in levels of 2-aminoadipate (*p*-value = 1.47 × 10^−19^) and 2-oxoadipate (*p*-value = 7.98 × 10^−36^), which are products of its catabolism [17]. The significant decrease observed in the levels of methionine and methionine sulfoxide, along with the increase in cysteine, cystathionine and taurine suggest enhancement in taurine synthesis during endurance racing (Figure 4). The significant decrease observed in the levels of arginine (*p*-value = 4.06 × 10^−23^), citruline (*p*-value = 9.74 × 10^−22^), and ornithine (*p*-value = 2.95 × 10^−18^), together with increase in homocitrulline (*p*-value = 5.03 × 10^−8^) levels, suggest an upsurge in the urea cycle metabolism.

Among the carbohydrates, we observed a significant decrease in glucose levels (*p*-value = 6.15 × 10^−11^) accompanied by a significant increase in 3-phosphoglycerate (*p*-value = 2.43 × 10^−7^), pyruvate (*p*-value = 1.37 × 10^−13^), and lactate (*p*-value = 1.57 × 10^−32^), which suggest the activation of anaerobic glycolysis under endurance racing conditions. The levels of other monosaccharides (fructose and mannose) were decreased whereas the levels of amino sugars (e.g., glucoronate, erythronate, and N-glycolylneuraminate) were elevated.

Taken together, multiple metabolic pathways are altered to support the physiological processes triggered by endurance racing. 

### 2.4. Endurance Racing Accelerates Clearance of Red Blood Cells as Depicted by Metabolomics 

Given that oxygen is required for utilization of fatty acids (β-oxidation), and that erythrocytes support oxygen transport, which is bound to hemoglobin, we hypothesized that endurance racing would affect erythrocytes and heme metabolism. Indeed, we observed a significant increase in the levels of heme (*p*-value = 1.69 × 10^−8^) and products of its catabolism, namely biliverdin (1.84 × 10^−11^) and bilirubin (2.62 × 10^−5^), which might suggest a disruption of red blood cells (hemolysis). As hemolysis results in a reddish coloring of the plasma, to further confirm that elevated heme metabolism could be associated with the red blood cell disruption, we investigated the color of the plasma before and after the race. The color of plasma before the race was yellow and indeed differed from the orange to reddish color of the plasma samples collected after the race (Appendix A). We graded the degree of plasma color change as follows: 0—yellow; 1–minimal orange; 2—strong orange; 3—red. We then preceded to investigate whether parameters, such as animal age, distance ran, reason for elimination or their average speed during the race, had an impact on the noticed color difference. The average speed of horses participating in the 80 km races when compared to those running the 100 km and 120 km races was significantly different (*p*-value < 0.05). We observed significantly higher levels of average speed in samples graded as 2 and 3 than in samples graded as 0 (Appendix A), suggesting that the higher average speed induced hemolysis. The equine age, on the other hand, showed no significant impact on hemolysis. The differences in plasma color were not affected by dehydration, as no significant differences in the osmolality levels before and after the race were observed (Appendix A). 

Taken together, the accelerated heme metabolism could reflect on the red blood cell disruption which is related to the speed of the animal at the endurance race. 

### 2.5. Feasibility of Using Metabolic Signatures as Predictors of Animal Readiness for Endurance Race

The significant changes in multiple metabolic pathways, which have been triggered by the endurance racing, might suggest that those horses that completed a race could have some inherent metabolic advantages over those that were eliminated. To investigate such a possibility, we tested for plasma metabolic differences before the race, between the horses that finished and those that were eliminated. 

Interestingly, already before the race, we found 76 metabolites, predominantly amino acids, lipids and xenobiotics, showing nominally significant differences between finishers and the eliminated animals (Appendix A). The elevated levels of molecules involved in the urea cycle metabolism, BCAA and AAA catabolism together with lower levels of lipids, including long chain fatty acids and monoxydroxy fatty acids, were observed in those horses finishing the race.

Next, we tested whether the equine metabolic composition prior to the race could serve as a predictor of animal performance in the competition. We analyzed the metabolic profiles before the race from all animals, including the horses who completed the race as well as those eliminated. For the selection of metabolites that exhibit “stable” differential intensities between the groups of horses finishing the race and the ones which were disqualified, random forest and glmnet.lasso with stability selection were used (see Methods section). In order to include the metabolites which only display stable differential expression, only the 50 pre-selected metabolites, as well as age and heart rate (Appendix A), were used for model building. Twenty percent of the data were randomly selected and set aside as a test set for model evaluation. The R-function random Forest [18] was used to compute 1000 trees. The R-function glm.lasso with stability selection was used [19]. A total of six metabolites (imidazole propionate, pipecolate, ethylmalonate, 2R, 3R-dihydroxybutyrate, β-hydroxy-isovalerate and unknown X-25455) were selected using the stable parameters maxQ = 15 and cutoff = 0.6. The resulting model had an area under the receiver operating characteristic (ROC) curve (AUC) of 0.92 (95%CI = 0.85–0.98) (Figure 5A). The levels of metabolites identified as potential predictors of animal readiness for endurance racing showed nominal differences between finishers and eliminated animals. The finishers were showing elevated levels of imidazole propionate, pipecolate, 2R, 3R-dihydroxybutyrate, and β-hydroxy- isovalerate as well as lower levels of ethylmalonate, and unknown X-25455 (Figure 5B).

Taken together, we observed that before the race there were nominally significant differences in the metabolic composition between the animals finishing the competition and those disqualified. Moreover, we demonstrate the feasibility of deploying metabolic signatures as predictors of equine readiness for endurance competition.

## 3. Discussion

For many horses, endurance racing is challenging and between 30 and 70% of the animals are eliminated due to health conditions, such as lameness or metabolic conditions [3]. However, current measures of biochemical and physiological parameters are insufficient to assess readiness of the animal for qualification into the race, emphasizing an important ethical consideration to participate in endurance events.

Here, we replicated findings reported previously in multiple studies of accelerated β-oxidation of fatty acids, in response to endurance training, conducted in both human [7,20] and equine [10,11,13,21] subjects. The lipid catabolism has energetic advantages over the metabolism of carbohydrates [22], therefore it is activated under enhanced energetic need. In contrast to previous studies, we have identified a larger spectrum of lipids and provided detailed description on 167 different lipid molecules, significantly altered under the race, including molecules involved in α- and ω- oxidation of fatty acids. The α-oxidation plays a role in the degradation of branch chain fatty acids and can occur only in peroxisomes [23]. The catabolism of fatty acids in the process of ω-oxidation was previously described as a rescue pathway for fatty acid disorders in humans [23], and was also linked with anti-inflammatory function [24]. Thus, increased level of molecules involved in ω-oxidation, observed in our study, might suggest β-oxidation overload and activation of ω-oxidation, to support the lipid catabolism and meet the energetic needs of the organism. 

Notably, as β-oxidation generates higher levels of oxygen radicals and thus oxidative stress in comparison to glycolysis [25], this could suggest the need to activate pathways that regulate oxidative stress when β-oxidation increases. For instance, products of histidine catabolism, such as carnosine and anserine, were shown to possess anti-oxidative capacity, and were identified as pH-buffering, anti-glycation and calcium signaling molecules [4]. Therefore, increased levels of carnosine and anserine, observed in our study, suggest their activation in response to the accelerated lipid catabolism and potential oxidative stress triggered by an accelerated β-oxidation. Moreover, taurine, has previously been reported as a molecule with antioxidative properties [26] supporting mitochondrial function [27] and, in our study, was also identified to have elevated levels. Such findings further illustrate the tight interplay between metabolic pathways supporting organism adaptation to endurance racing conditions. Hence, our study, in contrast to previous reports [11,13], highlights the importance of metabolic processes beyond pathways involved in energy generation, further suggesting the potential benefits of monitoring such parameters which modulate oxidative stress in endurance horses. 

In accordance with previous reports, we have also observed that endurance racing enhances catabolism of BCAA, increases urea cycle metabolism and causes accumulation of lactate [9,10,21]. BCAA metabolism is activated by such strenuous exercises, along with protein catabolism, contributing to energy generation [28]. The catabolism of BCAA, as well as other amino acids, leads to the production of toxic ammonia which is metabolized via the urea cycle and excreted in the form of urea with urine [22]. Thus, enhanced urea cycle metabolism under endurance training, observed by us and others [11], is activated in response to accelerated protein and amino acids degradation. The accumulation of lactate observed in our study, previously seen as a waste product of metabolism, associated with the exercise-induced muscle fatigue [29], could rather suggest adaptation to exercise by contribution to energy generation as well as stimulation of blood flow, in light of current evidence [30]. The summary of metabolic responses to endurance racing is provided in Figure 6A.

Our study has also revealed the potential metabolic advantages of those animals who completed the race, which could not be monitored with the standard clinical chemistry approach. Furthermore, we showed the feasibility of deploying metabolomics to predict the animal endurance capacity under desert conditions. Although, the identified metabolic signatures were only nominally significant, possibly due to the small sample size, and would require replication, the identified metabolites contributing to the prediction of the race outcome were distributed over the metabolic pathways identified as relevant for endurance racing (Figure 6B). The identified increased levels of BCAAs metabolic products (beta−hydroxyisovalerate and ethylamlonate), together with a lower level of omega-6 free fatty acids, at resting state, could be considered as key features for a horse’s ability to complete the race competition. BCAAs are crucial components of proteins, and were shown to improve cell proliferation and muscle recovery after exercise, as well as a decrease in exercise-induced muscle damage [11,13]. Supplementation with BCAAs in human athletes was shown to enhance exercise capacity and lipid catabolism during endurance training [31]. Thus, BCAAs supplementation could be considered as one of the strategies to improve animal performance. Moreover, previous studies have shown that omega-6 free fatty acids were associated with low-grade inflammation and oxidative stress [32]. Hence, lower levels of omega-6 free fatty acids in animals who finished the race further supports their potential metabolic advantage over the disqualified animals. Given that the endurance exercise resulted in oxidative stress and inflammation [33], it could be suggested that the disqualified animals that presented with higher levels of omega-3 fatty acids did not recover from previous training and/or races. The maintenance of low omega-6/omega-3 ratio was suggested as a potential strategy for reducing inflammation [34] and, for that reason, supplementing endurance racing horses with omega-3 fatty acids could be considered to reduce race-induced inflammation. Furthermore, elevated levels of omega-6 fatty acids might serve as a signature of animal recovery. The 2R,3R−dihydroxybutyrate, a product of threonine metabolism [35], could suggest that AAA might play a significant role in the organism metabolism exposed to endurance exercise. The imidazole propionate is a product of histidine metabolism, which we observed as significantly altered in response to the race. However, previous studies suggest that imidazole propionate is a product of microbiota, which impairs insulin signaling in type 2 diabetes [36]. Nevertheless, our study, in contrast to previous reports, was conducted in elite equine athletes highly sensitive to insulin, and thus the role of imidazole propionate in such a context would require further investigation. Lastly, pipecolate, the product of lysine metabolism, identified as a potential predictor of equine performance in endurance racing, was previously associated with the protein turnover in muscle fibers (myotubes) [37]. Therefore, it could be suggested that animals with higher pipecolate levels have metabolic advantages related to protein turnover in the muscles, which is a crucial component of endurance exercises [38].

Our study, however, also has several limitations that would require additional investigation. Firstly, it was conducted in a small sample size and future studies would require a bigger cohort. Secondly, we showed metabolic predictors of animals’ endurance capacity, however, our study requires further replication to verify these findings. Thirdly, we have used untargeted metabolomics profiling, which is suitable for discovery, however studies deploying targeted, quantitative assays will have to be developed for future implementation. Furthermore, a study testing the impact of the dietary supplements suggested in this manuscript, such as BCAAs, omega-3 fatty acids or histidine, on the performance capacity of endurance horses would be required.

In this study, we have characterized broad metabolic changes driven by endurance racing, identifying metabolic signatures, which could potentially be implemented to assess fitness of the animals to participate and qualify in endurance competitions. To the best of our knowledge, this study is the first to show the feasibility of using metabolomics to predict the endurance capacity of horses. The metabolites identified in our study can be further investigated in the context of animal physiology and optimization of performance capacity. 

## 4. Materials and Methods 

### 4.1. Animal and Competition Information

Horses participating in 3 different events, with at least 1 month apart, at the Qatar Endurance Village in Mesaieed, Qatar, were considered for the study. Consent forms were obtained for all of the horses included in the study. Initially, a total of 50 horses were sampled at least one time. The inclusion criteria used in this study were: 1) having the owner’s consent to participate in the study; 2) passing the initial vet check; 3) having blood samples and heart rate taken before and after exercise. Only 47 animals fulfilled all of these criteria and were included in the study. These 47 horses were Arabian or half-breed Arabian: 6 stallions, 19 geldings and 22 mares, with a mean age of 12 ± 3.3 years old. 

Depending on the competition distance, the races were divided into 3 to 4 compulsory loops with pauses for rest and for veterinarian assessment called vet checks. Horse inspections proceeded in accordance to Fédération Equestre Internationale (FEI) [16] rules by the designated official veterinarians, with assessment of the animals fitness to continue the competition based on its heart rate recovery, metabolic status, gait and general condition. Animals presented for inspection with a heart rate above 64 beats per minute after the regulated recovery time or that had significant cardio-respiratory or gastrointestinal alterations were eliminated due to metabolic conditions. Part of the clinical examination gait analysis of the animals is assessed at trot. When presented with an abnormal gait, the animal was eliminated due to lameness condition.

### 4.2. Study Design

The race distances in these 3 events were 80, 100 or 120 km, and the number of animals sampled for each distance was 13, 23 and 26, respectively. Some of the animals were racing in multiple competitions giving a total of 62 sampled animals (34 horses sampled in 1 race, 11 horses sampled in 2 races, and 2 horses sampled in 3 races).

Horses were sampled on two occasions during each race. First sample was collected at the time of the vet check on the afternoon preceding the race corresponding to the before race (BR) sample (nBR = 62). The second sample was taken within 30 min after the end of the race or upon elimination during the race corresponding to the after race (AR) sample (nAR = 62). For each horse, the heart rate (HR) BR and AR was recorded. These were obtained at the vet gate by the official veterinarians designated for the races. The race status (finisher, eliminated by lameness and metabolic disease or other) and the final average speed obtained from the final reports published by Qatar Endurance Committee.

This study design was approved by the Institutional Animal Care and Use Committee from Weill Cornell Medicine—Qatar under the number WCMQ-2018-003.

All races were divided into several sections, ranging from 20 to 40 km in length, separated by compulsory halts for vet gates followed by a mandatory period of recovery of 20 min. 

### 4.3. Sample Collection

With minimal restraint, blood was collected from a jugular vein and transferred into 6mL ethylenediaminetetraacetic acid (EDTA) commercial evacuated tubes. Samples were gently inverted 8–10 times to allow even distribution of the anticoagulant. Blood was refrigerated and brought to the field laboratory, set up for the purpose of the study. Time from blood collection and centrifugation was no longer than 30 min. Each sample was centrifuged for 10 min at 1500× *g* at room temperature. The obtained plasma was aliquoted into pre-cooled storage vials, with approximately 200 µL each. All samples were maintained at −80 °C until processed. A total of 124 samples were collected (62 BR and 62 AR).

### 4.4. Clinical Chemistry Measurements

All clinical chemistry measurements including albumin, alkaline phosphatase, aspartate transaminase, total bilirubin, creatine kinase, creatinine, gamma glutamyl transferase, lactate, lactate dehydrogenase, phosphatase, and blood urea nitrogen were conducted in diagnostic laboratory at Equine Veterinary Medical Center, Doha, Qatar. The EDTA plasma samples were measured using automated clinical chemistry Cobas C311 analyzer (Roche Diagnostics, Risch-Rotkreuz, Switzerland).

### 4.5. Metabolic Measurements 

All metabolomics measurements were conducted using the Metabolon Inc. HD4 platform, implemented at the Anti-Doping Lab in Qatar (ADLQ). Sample processing and metabolic profiling was conducted using an automated MicroLab STAR® (Hamilton, Reno, NV, USA) system as previously described [39,40]. Briefly, the samples were mixed with recovery standards and extracted using a methanol-based solvent. The obtained sample extracts were divided into equal parts, evaporated under nitrogen stream (TurboVap (Zymark)), and reconstituted in four different solvents compatible with each of the four analytical methods including: 1) acidic positive ion (optimized for hydrophilic compounds)—extract gradient eluted from a C18 column (Waters UPLC BEH C18–2.1 × 100 mm, 1.7 μm) with water and methanol containing 0.05% perfluoropentanoic acid and 0.1% formic acid; 2) acidic positive ion (optimized for hydrophobic compounds)—extract gradient eluted from C18 (Waters UPLC BEH C18–2.1 × 100 mm, 1.7 μm) with methanol, acetonitrile, water, 0.05% perfluoropentanoic acid, and 0.01% formic acid; 3) basic negative ion—extract gradient eluted from a separate dedicated C18 column using methanol and water containing 6.5 mM ammonium bicarbonate at pH 8; and 4) negative ionization—extract gradient eluted from a HILIC column (Waters UPLC BEH Amide 2.1 × 150 mm, 1.7 μm) using water and acetonitrile with 10 mM ammonium formate at pH 10.8 [41,42].

The measurements were conducted using Waters ACQUITY ultra-performance liquid chromatography (UPLC) and a Thermo Scientific Q-Exactive high resolution/accurate mass spectrometer interfaced with a heated electrospray ionization (HESI-II) source and Orbitrap mass analyzer operated at 35,000 mass resolution [41]. The raw data were submitted to Metabolon Inc. (Durham, NC, USA) for compound identification deploying Metabolon’s hardware and software. The components’ identification was conducted by comparison of peaks to library entries of purified standards based on retention index, accurate mass match to the library ± 10 ppm, and MS/MS forward and reverse scores between the experimental data and authentic standards [40]. The obtained metabolic data were normalized to correct for variations resulting from inter-day tuning differences in the instrument. Each compound was corrected in a run day. Instrument variability, 7%, was determined by calculating the median relative standard deviation (RSD) for the standards that were added to each sample prior to injection into the mass spectrometers. Overall process variability was determined by calculating the median RSD for all endogenous metabolites (i.e., non-instrument standards), 11%, present in 100% of the pooled matrix samples. 

### 4.6. Osmolality Measurements

Osmolality measurements were conducted at ADLQ using a freezing-point Fiske Micro-Osmometer Model 210 (Norwood, MA, USA). Osmolality was determined using 20 µL of sample, and measurements were performed in duplicate. 

### 4.7. Statistical Analysis

Data analysis was performed using R studio (R version 3.5.3) and R-package limma50 (25) through our in-house developed tool “autonomics” (freely available at https://github.com/bhagwataditya/autonomics). Metabolite levels were scaled by run-day medians and log-transformed. Then, we fitted general linear models METABOLITE~ (1 + BEFOREAFTER + FINISHED + SEX + AGE), where BEFOREAFTER, FINISHED and SEX are coded as dichotomous variables, respectively, representing the time point (before/after the race), whether the horse finished the race (yes/no) and sex of the animal (male/female). The horse age was coded in years, and investigated the following contrast: ("After the race–Before the race"). Then, to test for the metabolic differences between finishers and eliminated animals, we limited the data set to samples collected before the race and we fitted a linear model METABOLITE ~ 1+ FINISHED + SEX + AGE + HR (heart rate), and investigated the following contrast: (“Disqualified - Finishers”). 

Finally, for the assessment of metabolites as potential predictors of competition outcome, we pre-selected the 50 most informative metabolites, using random forest classification, to discriminate race outcome (finish yes/no). We then used these 50 metabolites, along with age and heart beat rate information in a second step, where we used glmnet.lasso regression with stability selection (23) to identify a model with a maximum of 15 predictors (stabsel parameter maxQ) to represent a compromise between simplicity of the model (to avoid over-fitting) and predictive power. The R-function random Forest (22) was used to compute 1000 trees. The stable parameters maxQ = 15 and cutoff = 0.6 was used

## Figures and Tables

**Figure 1 metabolites-11-00082-f001:**
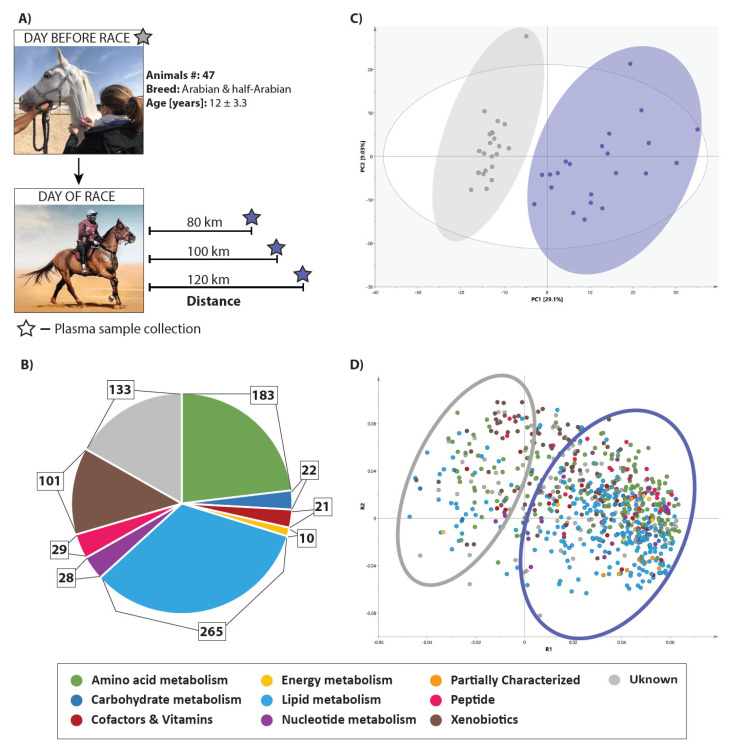
Endurance race triggers global changes in equine metabolism. (**A**) Overview on the experimental design. (**B**) Pie chart reflective of the number of metabolites measured. (**C**) The principal component analysis (PCA) of metabolites measured in equine plasma before and after the race. (**D**) PCA loading plot shows contribution of different metabolic classes to the separation. Grey—before the race; Blue—after the race.

**Figure 2 metabolites-11-00082-f002:**
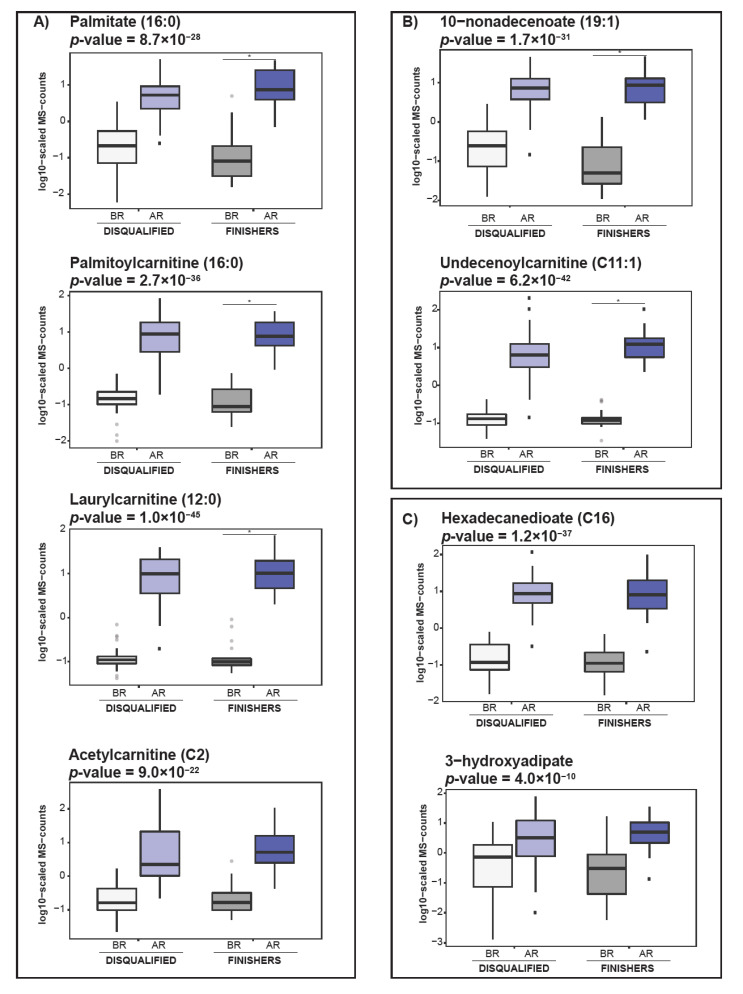
Lipid catabolism is accelerated in response to endurance racing. (**A–C**) Box plots showing examples of alterations in molecules involved in various lipid catabolism pathways. (**A**) Represents molecules involved in β-oxidation. (**B**) Represents molecules involved in α-oxidation. and (**C**) Represents molecules involved in ω-oxidation. Light grey—disqualified animals before the race; dark grey—finishers before the race; light blue—disqualified animals after the race; dark blue—finishers after the race. “*”—indicate significant changes.

**Figure 3 metabolites-11-00082-f003:**
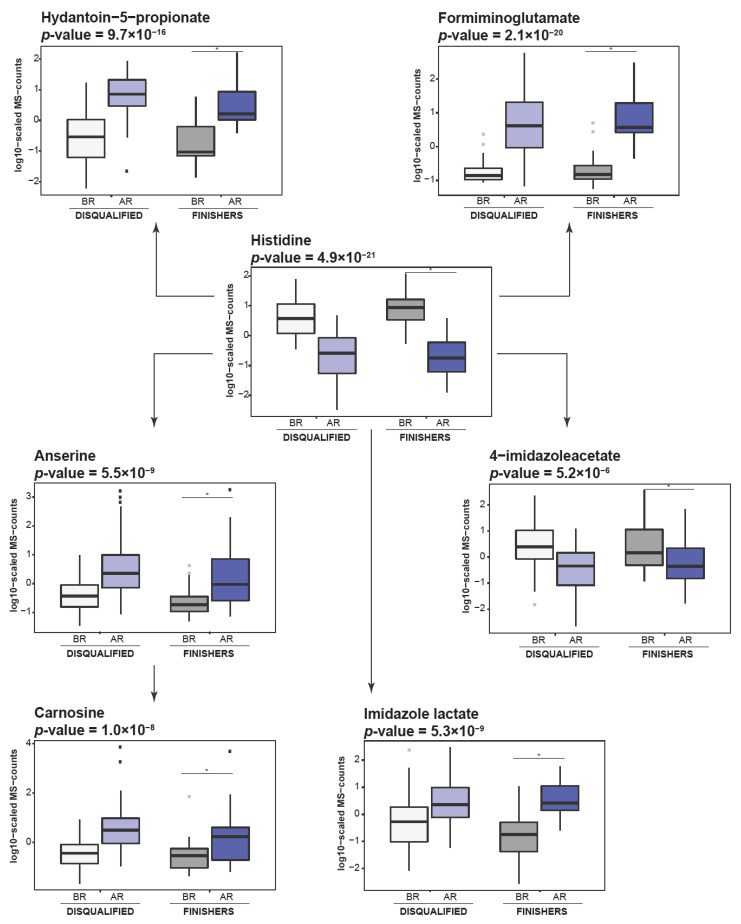
Endurance race induces histidine catabolism. Box are plots showing metabolites involved in catabolic pathway of histidine. Light grey—disqualified animals before the race; dark grey—finishers before the race; light blue—disqualified animals after the race; dark blue—finishers after the race. “*”—indicate significant changes.

**Figure 4 metabolites-11-00082-f004:**
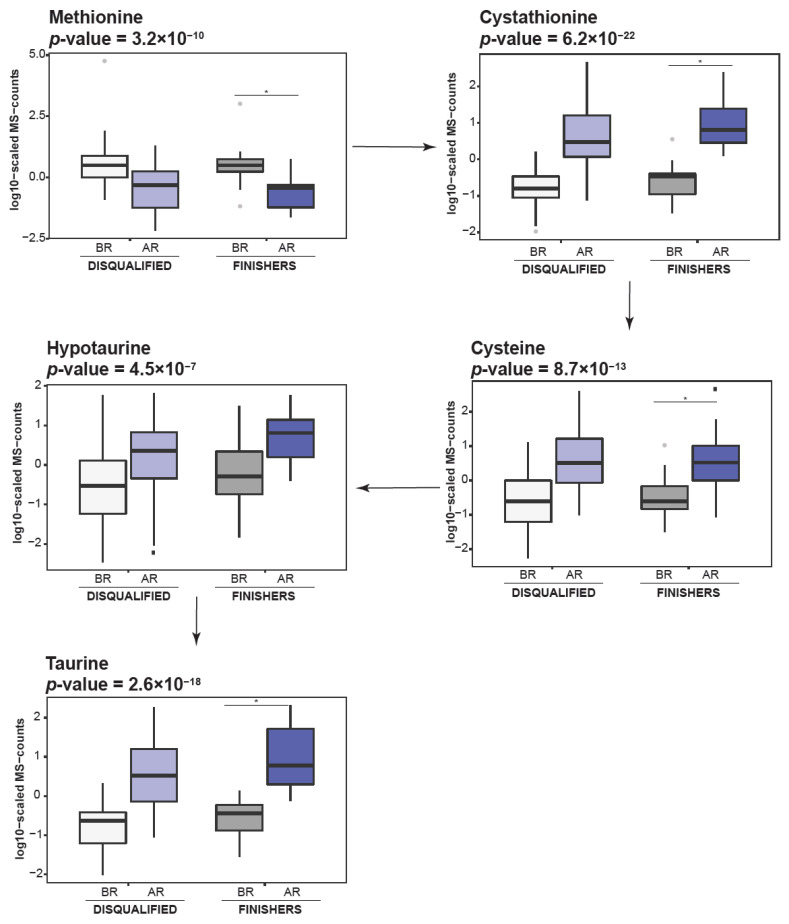
Taurine synthesis is enhanced under the endurance race. Box plots are showing examples metabolic intermediates in taurine synthesis. Light grey—disqualified animals before the race; dark grey—finishers before the race; light blue—disqualified animals after the race; dark blue—finishers after the race. “*”—indicate significant changes.

**Figure 5 metabolites-11-00082-f005:**
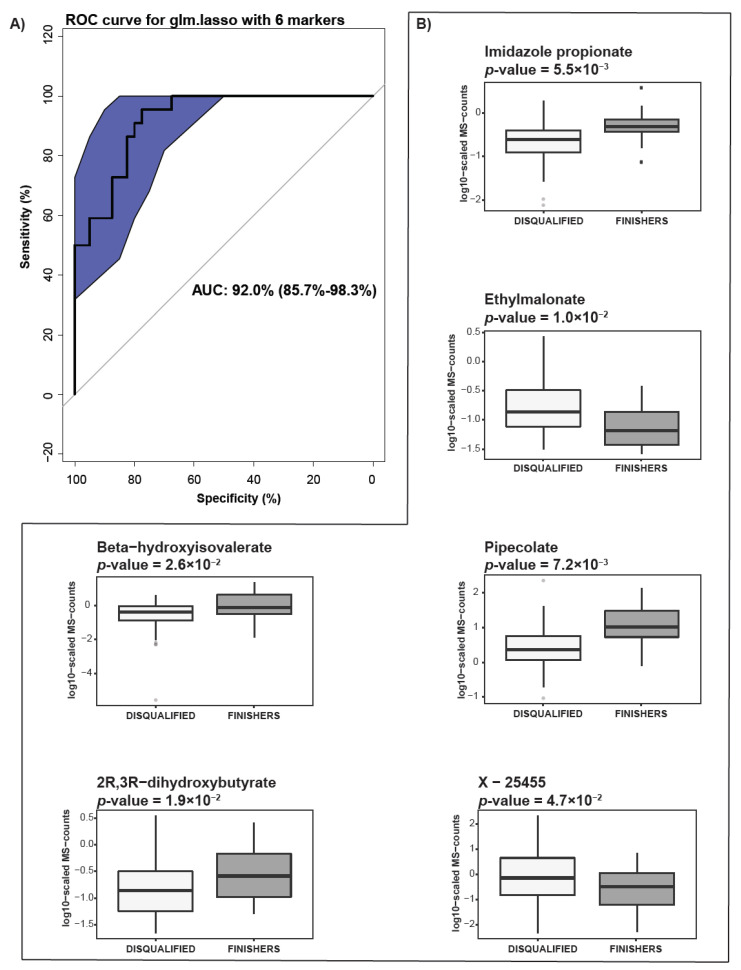
Feasibility of using metabolic signatures as predictors of animal readiness for endurance race. (**A**) Receiver operating characteristic (ROC) curve for glm.lasso with 6 markers. (**B**) Box-plots showing differences between finishers and disqualified animal in 6 metabolites identified by glm.lasso, used to create the ROC curve.

**Figure 6 metabolites-11-00082-f006:**
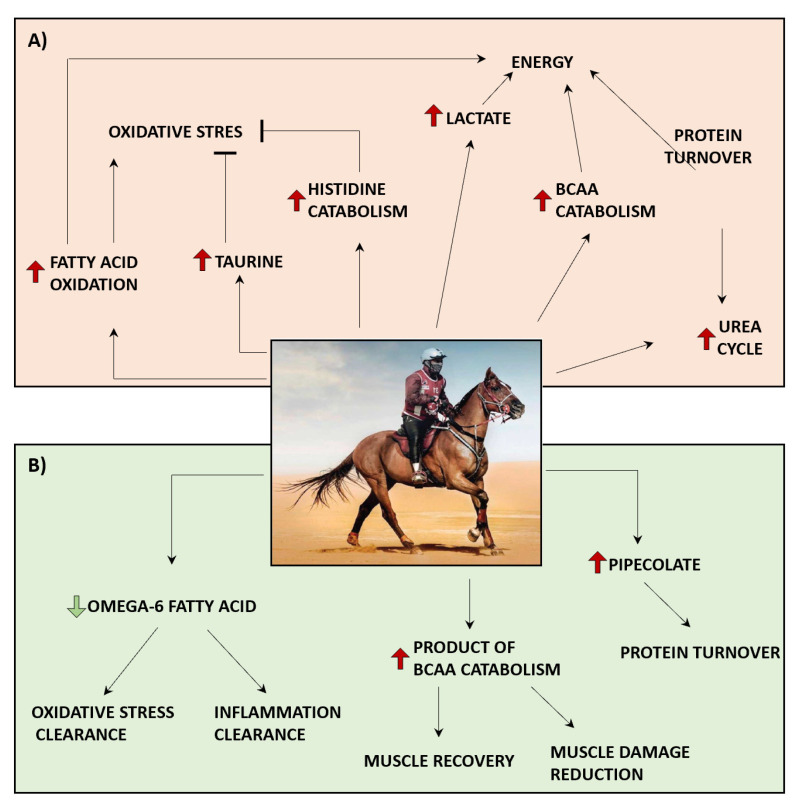
Metabolic processes in endurance equine. (**A**) Metabolic consequences of endurance racing (depicted in orange). (**B**) Metabolic features before the race associated with the animal readiness to participate in endurance competition (depicted in green). “↑”—stimulation; “⊣” inhibition; ↑—increase; ↓—decrease.

**Table 1 metabolites-11-00082-t001:** Animal characteristics and clinical chemistry results for animals before and after the race.

	Average [± S.D.]				
	All	Finishers	Lameness	Metabolic	*p*-Value
Age	11.79 [3.36]	11.82 [3.28]	11.55 [3.11]	11.67 [3.68]	NS
HR BR	50.02 [13.50]	38.96 [6.30]	42.23 [6.81]	39.73 [6.45]	NS
HR AR	60.23 [11.04]	54.09 [5.02]	57.64 [6.98]	73.07 [11.91]	1.18 × 10^−8^
ALB BR	33.54 [2.37]	34.31 [2.18]	32.60 [2.33]	33.61 [2.27]	NA
ALB AR	38.90 [ 4.60]	38.84 [3.96]	36.87 [4.35]	41.67 [4.39]	5.90 × 10^−3^
ALP BR	6.92 [3.19]	7.50 [3.06]	5.90 [3.34]	7.38 [2.85]	NA
ALP AR	8.26 [8.43]	6.42 [3.72]	9.38 [11.28]	9.56 [8.65]	NA
ALT BR	11.16 [8.55]	9.88 [4.07]	9.73 [4.32]	14.96 [14.49]	NA
ALT AR	27.82 [29.25]	26.83 [26.10]	24.49 [32.37]	33.68 [28.54]	NA
AST BR	339.49 [166.01]	304.33 [54.42]	330.43 [106.68]	404.13 [282.05]	NA
AST AR	532.31 [391.83]	493.48 [321.54]	500.46 [363.79]	632.38 [492.88]	NA
BIL BR	23.63 [9.78]	22.23 [9.73]	22.14 [7.38]	27.69 [11.35]	NA
BIL AR	43.71 [15.30]	42.47 [12.86]	38.00 [11.80]	53.06 [18.15]	9.49 × 10^−3^
CREA BR	105.72 [19.01]	104.88 [20.97]	105.81 [21.20]	106.88 [11.25]	NA
CREA AR	152.89 [39.96]	145.63 [30.14]	145.62 [43.88]	173.31 [40.40]	NA
GGT BR	19.25 [6.85]	19.50 [5.82]	18.52 [6.09]	19.81 [8.86]	NA
GGT AR	21.93 [10.38]	23.58 [13.36]	18.95 [5.76]	23.38 [9.04]	NA
GLU BR	6.25 [1.15]	6.48 [1.35]	6.10 [1.05]	6.11 [0.87]	NA
GLU AR	6.21 [2.08]	5.77 [2.05]	6.87 [2.15]	6.01 [1.78]	NA
LACT BR	0.67 [0.58]	0.81 [0.89]	0.59 [0.16]	0.57 [0.13]	NA
LACT AR	3.19 [2.24]	2.87 [0.91]	2.27 [1.56]	4.87 [3.26]	9.43 × 10^−4^
LDH BR	497.44 [145.61]	533.17 [106.41]	486.81 [137.68]	457.81 [187.97]	NA
LDH AR	876.66 [394.24]	930.33 [327.54]	692.33 [318.54]	1038.06 [472.76]	1.97 × 10^−2^
PHOS BR	0.86 [0.19]	0.83 [0.18]	0.89 [0.20]	0.87 [0.18]	NA
PHOS AR	1.20 [0.43]	1.25 [0.46]	1.09 [0.34]	1.26 [0.48]	NA
BUN BR	4.99 [1.02]	5.18 [1.02]	5.13 [1.03]	4.53 [1.87]	NA
BUN AR	7.74 [1.60]	8.44 [1.09]	7.09 [0.77]	7.56 [1.50]	1.42 × 10^−2^
TP BR	119.21 [38.48]	126.21 [43.47]	113.43 [32.44]	116.31 [36.12]	NA
TP AR	108.16 [20.51]	106.83 [21.08]	104.19 [21.06]	115.38 [16.74]	NA

HR—heart rate, BR—before the race, AR—after the race, ALB—albumin; ALP—alkaline phosphatase; AST—aspartate transaminase; BIL—total bilirubin; CREA—creatinine; GGT—gamma glutamyl transferase; GLU—glucose; LACT—lactate; LDH—lactate dehydrogenase; PHOS—phosphatase; BUN—blood urea nitrogen. NS—not significant. NA—not applicable.

## Data Availability

The data presented in this study are available on request from the corresponding author.

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
