# Peer review of "Metabolic Predictors of Equine Performance in Endurance Racing"

_metabolites, 2021, doi:10.3390/metabo11020082_

Round 1
Reviewer 1 Report
Minor revision (text editing) should be necessary.
Author Response
Minor revision (text editing) should be necessary.
Response: We thank the reviewer for this positive appreciation. The manuscript underwent extensive editing, which improves the manuscript quality.
Reviewer 2 Report
In the reality, the idea of the research is good. I do not prefer non targeted study. However, the approach is OK. The article is very confusing and requires major extensive revision. Authors claim that there were 6 predictors of animal performance in endurance competition (Where is it???). It is not mentioned even in the abstract section. Introduction is very long. Results section is the main confusing part. Normally no discussion of the results section and therefore no reference could be appear. it is recommended to illustrate the significance of main metabolites in comparison to the others. in discussion section you may explain the reasons that stand behind. along of the article; the abbreviations are very confusing like AAA; BCAA....etc. It is not real to say significant difference; you have to tell us significant increase or decrease. The figures are not conclusive for example Figure 1 is of no benefit. Please focus on the most pronounced markers (if present) and discuss its further use for targeted studies. The issue of non significant osmolarity is not clear
Author Response
In the reality, the idea of the research is good. I do not prefer non targeted study. However, the approach is OK.
Response: We thank the reviewer for this positive appreciation. We agree with the reviewer that untargeted metabolomics is the most suitable approach for the discovery study, which we have conducted, but cannot be applied for quantification.
The article is very confusing and requires major extensive revision. Authors claim that there were 6 predictors of animal performance in endurance competition (Where is it???). It is not mentioned even in the abstract section.
Response: Thank you for the comment. We have included the information on identified metabolic predictors in Abstract as well as having amended the description in Results section paragraph “Feasibility of using metabolic signatures as predictors of animal readiness for endurance race”.
Changes in Abstract [L25-L45]
“Further, even before the race starts, we found metabolic differences between animals who completed the race and those who didn’t. We identified a set of six metabolite predictors (imidazole propionate, pipecolate, ethylmalonate, 2R, 3R-dihydroxybutyrate, β-hydroxy-isovalerate and X-25455) of animal performance in endurance competition; the resulting model had an area under a receiver operating characteristic (AUC) of 0.92 [95% CI: 0.85 – 0.98].”
“The resulting model had an area under the receiver operating characteristic (ROC) curve (AUC) of 0.92 [95%CI = 0.85 – 0.98] (Figure 5 A). The levels of metabolites identified as potential predictors of animal readiness for the endurance race showed nominal differences between finishers and eliminated animals. The finishers were showing elevated levels of imidazole propionate, pipecolate, 2R,3R-dihydroxybutyrate, and β-hydroxy- isovalerate as well as lower levels of ethylmalonate, and unknown X-25455 (Figure 5B).”
Introduction is very long.
Response: Thank you for the comment. We agree with the reviewer that the introduction is a bit lengthy. We have conducted changes to improve this part of the manuscript. Please find the amended introduction part below.
Changes in Introduction lines: [L55-L70; L80-L115]
“Endurance sport refers to any kind of competition in which participants stamina is tested to its limits. The performance of endurance athletes depends on multiple components, including genetic predisposition, physiological, biomechanical, and psychological factors [1]. The endurance performance capacity could be maximized over the training, enabling to utilize the individuals genetic potential [2].
Endurance horses experience similar levels of physiological and metabolic stress as human athletes, and thus could be considered as a relevant in vivo model for the monitoring and establishment of a metabolic performance profile. For many horses, endurance racing is challenging and between 30 to 70% of animals are eliminated due to various health conditions, such as lameness and metabolic disturbances, in addition to environmental factors, riders ability and the horse’s level of experience [3]. Thus, strategies which could estimate the animal’s readiness to participate in endurance racing would be of great value, as this may reduce the number of unprepared animals, thus lowering their risk of developing certain health complications. However, currently there is no method that could estimate the equine endurance capacity.
The technological advancement in mass spectrometry (MS) contributed to the development of metabolomics, which offers detailed description of the small molecule (metabolite) composition of the organism [4]. Implementation of metabolomics approaches, in the field of endurance exercises resulted in comprehensive characterization of altered metabolic processes as a result of undergoing endurance performance [4–7]. Metabolic shift from carbohydrate towards lipid catabolism was suggested as the main energy source under the endurance challenge in humans [5,6] as well as equine [8–11]. Additionally, amino acid catabolism and hemoglobin metabolism [5,12], as well as oxidative stress [6], were found to be triggered by such exercises. Metabolic pathways that are predominantly affected by endurance training could be further analyzed to define individual metabolic fitness and capacity to endurance sport exposure. Nevertheless, metabolic signatures indicating performance capacity and conditioning for endurance training remain elusive.
Our aim here was to provide an in-depth characterization of metabolic consequences of endurance racing in Arabian and half-Arabian breeds. Furthermore, we aimed to establish a metabolic performance profile for those animals which could serve as a proxy for the animal’s readiness to participate in an endurance race. To achieve this, we monitored blood plasma samples of selected animals before and after completing endurance competition, as well those eliminated due to lameness or metabolic conditions, using an MS- based broad untargeted metabolic profiling method. Noteworthy, previous studies have so far only been performed in moderate temperate regions and only using more limited NMR methods for metabolic measurements [11,13]. Our study is the first of its kind to be performed under desert conditions and deploying broad untargeted metabolic profiling.”
Results section is the main confusing part. Normally no discussion of the results section and therefore no reference could be appear.
Response: Thank you for the comment and suggestion. The references in the results section were included to ensure that the research will be placed in the correct context. Given that this manuscript is linking two different fields of expertise, namely veterinary and metabolomics, we think that by referring to the previous study or processes, which are already known, in one field but not in other we will be inclusive for different readers.
It is recommended to illustrate the significance of main metabolites in comparison to the others. In discussion section you may explain the reasons that stand behind. along of the article.
Response: Thank you for the comment and suggestion. The metabolites and metabolic pathways which were showing the most prominent alterations in response to endurance race were visualized in form of bar plots in Figure 2 – 4. The p-value defining the level of significance was also included in the graph. The six metabolites which we have identified as metabolic predictors of equine performance capacity were included in Figure 5 in the form of box plots. We also prepared a new Figure 6 where the metabolic responses to endurance race were summarized and the metabolic predictors of animal performance were placed in the physiological context.
Change in Figures: Figure 6 added: [L581-L586]
Figure 6. Metabolic processes in endurance equine. A) Metabolic consequences of endurance race (depicted in orange). B) Metabolic features before the race associated with the animal readiness to endurance competition (depicted in green). ↑ - stimulation; - inhibition; ↑ - increase; ↓ - decrease.
The abbreviations are very confusing like AAA; BCAA....etc.
Response: Thank you for the comment. We have provided the explanation for abbreviation in the text.
“….molecules involved in branch chain amino acid (BCAA), aromatic amino acid (AAA),”
It is not real to say significant difference; you have to tell us significant increase or decrease.
Response: Thank you for the comment. The whole manuscript was checked to ensure that information on the directionality of the changes will be included. The changes were conducted according with the suggestion. Please see the conducted amendments below:
Changes in Results line: [L145-L147; L286-L291]
“After the race, we observed significant increase in the heart rate as well as levels of albumin (ALB), total bilirubin (BIL), lactate (LACT), and lactate dehydrogenase (LDH), along with significant decrease in blood urea nitrogen (BUN) between the animals completing the race and those eliminated (Table 1).”
“The levels of metabolites identified as potential predictors of animal readiness for the endurance race showed nominal differences between finishers and eliminated animals. The finishers were showing elevated levels of imidazole propionate, pipecolate, 2R, 3R-dihydroxybutyrate, and β-hydroxy- isovalerate as well as lower levels of ethylmalonate, and unknown X-25455 (Figure 5B).”
The figures are not conclusive for example Figure 1 is of no benefit.
Response: Thank you for the comment. The Figure 1 is providing an A) overview on the experimental design; B) number of measured metabolites and their distribution across metabolic classes; as well as C) & D) are showing how the endurance race is impacting the metabolism.
To further highlight the importance and relevance of all figure components, we conducted amendments in the body text. Please see the changes below:
Changes in Results line: [L128; L172-173; L178-181]
“The experimental design is presented in Figure 1 A.” --> this sentence was moved to the front of results section.
“We quantified relative levels of 792 metabolites, including 659 molecules of known identity, and 133 molecules of unknown identity. The distribution of measured metabolites across different metabolic classes is presented in Figure 1 B and all measured metabolites are listed in Supplementary Table 1.”
“Principal component analysis (PCA) was conducted on the metabolite levels identified from those horses that were not eliminated and finished the competition. We found a clear separation into two groups between before and after the race (Figure 1 C) suggesting metabolic alterations in response to the endurance competition.”
Please focus on the most pronounced markers (if present) and discuss its further use for targeted studies.
Response: Thank you for the comment. We regret that the paragraph in discussion section was not clear. To provide more clarity, we have intensively worked on the discussion to avoid the redundancy and highlight the most pronounced markers.
Changes in discussion line: [L562-575; L609-L614]
“Our study has also revealed the potential metabolic advantages of those animals who completed the race, which could not be monitored with the standard clinical chemistry approach. Furthermore, we showed the feasibility of deploying metabolomics to predict the animal endurance capacity under desert conditions. Although, the identified metabolic signatures were only nominally significant, possibly due to the small sample size, and would require replication, the identified metabolites contributing to the prediction of the race outcome, are distributed over the metabolic pathways identified as relevant for endurance racing (Figure 6B). The identified increased levels of BCAAs metabolic products (beta−hydroxyisovalerate and ethylamlonate), together with a lower level of omega-6 free fatty acids, at resting state, could be considered as key features for horse ability to complete the race competition. BCAAs are crucial components of proteins, and were shown to improve cell proliferation and muscle recovery after exercise, as well as a decrease in exercise-induced muscle damage [11,13]. Supplementation with BCAAs in human athletes was shown to enhance exercise capacity and lipid catabolism during endurance training [32].”
“Thirdly, we have used untargeted metabolomics profiling, which is suitable for discovery, however studies deploying targeted, quantitative assays will have to be developed for future implementation. Furthermore, a study testing the impact of the dietary supplements suggested in this manuscript, such as BCAAs, omega-3 fatty acids or histidine, on the performance capacity of endurance horses would be required.”
The issue of non significant osmolarity is not clear.
Response: Thank you for the comment. We agree that the paragraph reflecting on the osmolality was confusing. The following changes were conducted to improve:
Change in Results line: [L313-L319]
“We also tested for dehydration, which could affect animal performance and result in elimination. We measured plasma osmolality, as a proxy for animal dehydration [14]. There were no significant differences in the osmolality levels between finishers and eliminated animals before the race (Supplementary Figure 1B) as well as after the race (Supplementary Figure 1C), suggesting that the performance was not affected by the animal hydration. Additionally, no significant changes in the osmolality levels before and after the race were observed across animals participating in all distance (80 km, 100 km, 120 km) recess (Supplementary Figure 1D)”
Reviewer 3 Report
Revision of the manuscript „Establishing a metabolic performance profile for endurance race horses”.
The manuscript is interesting and well written, however, not all is clear. The main problem is the lack of information when the horses were eliminated from the competition. Which distance was covered by these horses before elimination?
This information is important for further analysis of the results. The longer distance the horses covered, the more tired they were and the greater the changes in the level of the parameters tested.
Some other comments:
Please, give the meaning of the used abbreviations in line 35 – AUC, in line 249 – BCAA, AAA
Line 95 “enhances glycogen transport” – what does it mean? Maybe glycogen turnover rate?
Lines 381-401 - In my opinion, this paragraph adds nothing new and may be omitted without prejudice to the work as a whole, especially, that the text is very long.
Author Response
The manuscript is interesting and well written, however, not all is clear.
Response: We thank the reviewer for this positive appreciation.
The main problem is the lack of information when the horses were eliminated from the competition. Which distance was covered by these horses before elimination? This information is important for further analysis of the results. The longer distance the horses covered, the more tired they were and the greater the changes in the level of the parameters tested.
Response: Thank you for the comment and suggestion. The information on the actual distance achieved by each animal was included in Supplementary Table 1. For the analysis of metabolic responses to endurance race we have used the metabolic data obtained only from the animals finishing the competition. To clarify we have amended the following:
Changes in Results line: [L178-L181; L192; L268]
““Principal component analysis (PCA) was conducted on the metabolite levels identified from those horses that were not eliminated and finished the competition. We found a clear separation into two groups between before and after the race (Figure 1 C) suggesting metabolic alterations in response to the endurance competition.”
“In those animals that finished the race we identified 417 metabolites, which were significantly altered after the race at a stringent Bonferroni level of significance (correction for number of metabolites (0.05/792), p-value = 6.31x10-5) (Supplementary Table 2).”
To determine metabolic predictors of animal readiness for the competition we have used only the metabolic measurements conducted before the endurance race. To clarify, we have included following:
“Interestingly, already before the race, we found 76 metabolites, predominantly amino acids, lipids and xenobiotics, showing nominally significant differences between finishers and the eliminated animals (Supplementary Table 3).”
Please, give the meaning of the used abbreviations in line 35 – AUC, in line 249 – BCAA, AAA
Response: Thank you for the comment. The abbreviations were included:
“…the resulting model had an area under a receiver operating characteristic (AUC)…”
“...branch chain amino acid (BCAA), aromatic amino acid (AAA)...”
Line 95 “enhances glycogen transport” – what does it mean? Maybe glycogen turnover rate?
Response: Thank you for the comment. We are sorry for the confusion. Indeed, we meant glycogen turnover. However, the introduction underwent intensive modification, and the sentence was removed to shorten the introduction.
Lines 381-401 - In my opinion, this paragraph adds nothing new and may be omitted without prejudice to the work as a whole, especially, that the text is very long.
Response: Thank you for the comment. We agree with the reviewer that the text could be omitted. We have removed this paragraph. We have also worked on the other components of the discussion to avoid redundancy.
Changes in Discussion line: [L303-L318]
Round 2
Reviewer 2 Report
The authors respond successfully to all of my Comments. Thanks
This manuscript is a resubmission of an earlier submission. The following is a list of the peer review reports and author responses from that submission.